# Detection of Genomic Structural Variations Associated with Drug Sensitivity and Resistance in Acute Leukemia

**DOI:** 10.3390/cancers16020418

**Published:** 2024-01-18

**Authors:** Darren Finlay, Rabi Murad, Karl Hong, Joyce Lee, Andy Wing Chun Pang, Chi-Yu Lai, Benjamin Clifford, Carol Burian, James Mason, Alex R. Hastie, Jun Yin, Kristiina Vuori

**Affiliations:** 1NCI-Designated Cancer Center, Sanford Burnham Prebys Medical Discovery Institute, La Jolla, CA 92037, USA; rmurad@sbpdiscovery.org (R.M.);; 2Bionano Genomics Inc., San Diego, CA 92121, USA; 3Scripps MD Anderson, La Jolla, CA 92037, USA

**Keywords:** leukemia, structural variant, drug screening, optical genome mapping, chemovulnerabilities

## Abstract

**Simple Summary:**

Whilst association of genetic mutations with targeted therapies is common in leukemia, to our knowledge, no one has associated genomic structural variants with drug sensitivities. Here we use optical genome mapping as an unbiased, genome-wide detection method for structural variants and show that some of these events are associated with sensitivity or resistance to clinically relevant anti-cancer drugs.

**Abstract:**

Acute leukemia is a particularly problematic collection of hematological cancers, and, while somewhat rare, the survival rate of patients is typically abysmal without bone marrow transplantation. Furthermore, traditional chemotherapies used as standard-of-care for patients cause significant side effects. Understanding the evolution of leukemia to identify novel targets and, therefore, drug treatment regimens is a significant medical need. Genomic rearrangements and other structural variations (SVs) have long been known to be causative and pathogenic in multiple types of cancer, including leukemia. These SVs may be involved in cancer initiation, progression, clonal evolution, and drug resistance, and a better understanding of SVs from individual patients may help guide therapeutic options. Here, we show the utilization of optical genome mapping (OGM) to detect known and novel SVs in the samples of patients with leukemia. Importantly, this technology provides an unprecedented level of granularity and quantitation unavailable to other current techniques and allows for the unbiased detection of novel SVs, which may be relevant to disease pathogenesis and/or drug resistance. Coupled with the chemosensitivities of these samples to FDA-approved oncology drugs, we show how an impartial integrative analysis of these diverse datasets can be used to associate the detected genomic rearrangements with multiple drug sensitivity profiles. Indeed, an insertion in the gene *MUSK* is shown to be associated with increased sensitivity to the clinically relevant agent Idarubicin, while partial tandem duplication events in the *KMT2A* gene are related to the efficacy of another frontline treatment, Cytarabine.

## 1. Introduction

Leukemias, including acute myeloid leukemia (AML, the main focus of this study), are diverse hematological cancers characterized by the bone marrow’s excessive production of certain immature myeloid blood cell subtypes. The American Cancer Society estimates that there will be ~59,610 new incidences of leukemia in the U.S. in 2023, with a prediction of ~23,710 deaths. Specifically for AML, ~20,380 new cases are predicted, with an estimate of ~11,310 deaths (https://www.cancer.org/cancer/types/acute-myeloid-leukemia/about/key-statistics.html, accessed 16 August 2023). Although there are multiple subtypes, with differing prognoses and defined treatments, more than 30% of patients will fail to enter complete remission upon currently accepted chemotherapy regimens, and almost all patients will eventually relapse.

The current standard of care (SoC) for AML has been the same for decades. Various combinations of anthracyclines and nucleoside analogues are used as frontline agents but have debilitating toxicities. Furthermore, most patients that respond to treatment initially but are not eligible for bone marrow transplantation will relapse within 5 years. As such, the need for novel targets and drugs to treat this disease is of paramount importance. This is especially true in patient populations unable to tolerate aggressive chemotherapy or in cases where bone marrow transplantation is not recommended or available. Whilst the identification of genetic mutations in hematopoietic stem and progenitor cells has been described, these observations have not resulted in successful personalized or precision interventions in AML. Indeed, studies have reported only approximately one to eight mutations per case in AML (e.g., [1,2,3]), yet there still is an observed heterogeneity of around fifteen disease subtypes. Initial observations of genomic structural variants (SV) in different forms of leukemia using high-resolution methods have captured roughly 40–80 rare SVs per individual AML patient [4], suggesting that SVs occur at a relatively higher rate and may help explain pathogenesis and disease heterogeneity.

In this study, genomic SVs were directly detected in leukemia samples via optical genome mapping (OGM) using the Bionano Saphyr system. This system has been used and validated extensively by multiple groups, spanning diverse applications such as genome assembly [5], facioscapulohumeral muscular dystrophy (FSHD) [6,7], Duchenne muscular dystrophy (DMD) [8], and SV detection in chimpanzee (*Pan troglodytes*) sub-populations [9,10]. Furthermore, relevant to this study, the platform has also been utilized to uncover novel variants in diverse types of cancer [11], including prostate cancer [12], liposarcoma [13], and multiple leukemia indications [4,14,15,16,17,18,19,20,21,22]. This technique can detect SVs at a ~5% allele fraction, thus allowing for the inference of clonal populations with distinct SVs [20].

Here, we describe a study aiming to associate chemical sensitivity screening with somatic SVs using leukemia cells directly isolated from individual patients. Traditionally, targeted agents are selected based on the presence of genomic mutations, but, to our knowledge, the association of SVs with drug sensitivities has not yet been applied. Thus, our rationale was to leverage this novel SV dataset with our clinically relevant drug screening data in an unbiased integrative analysis that could potentially result in further patient sub-stratification and/or drug repurposing. We assessed cell viability in response to treatment with 120 FDA-approved and late-stage investigational oncology drugs with the ultimate goal of identifying new drugs and targets to treat sub-populations of patients with leukemia or individual patients in the form of precision medicine. The integration of SVs and chemical sensitivities for unbiased statistical analyses reveals multiple novel potentially causative or pathogenic SVs, many of which are statistically associated with drug efficacy and resistance.

## 2. Methods

### 2.1. Primary Leukemia Cell Samples

Samples from patients with leukemia were obtained from Scripps MD Anderson (La Jolla, CA, USA) under the approved IRB protocol 13-6180, with each participant providing informed written consent. Briefly, blood samples were obtained from the central catheter, a peripheral blood draw, or a PICC line, while marrow aspirates were occasionally available if excess material was obtained during other procedures. The samples were processed within three hours of being obtained from the patients. The centrifugation of the samples through Ficoll-Paque^®^ PLUS (17-1440-02, GE Healthcare, Chicago, IL, USA), followed by residual red blood cells’ lysis (Alfa Aesar, cat. #J62990, Haverhill, MA, USA), resulted in enriched populations of nucleated cells. These were retained as aliquots in a Bambanker cell-freezing medium (Wako Pure Chemical Industries, Ltd., Neuss, Germany) on the day of receipt and kept in liquid N_2_ until use.

### 2.2. Ultra-High Molecular Weight DNA Isolation and Optical Genome Mapping

Optical genome mapping was carried out using the Saphyr system (Bionano Genomics Inc., San Diego, CA, USA) exactly as per the manufacturer’s instructions. Ultra-high molecular weight (UHMW) DNA was extracted from 1.5 M frozen cells using the Bionano Prep SP Blood and Cell Culture DNA Isolation Kit (#80030). The quantification of the DNA was performed with the Broad Range assay kit and read on a Qubit 2.0 Fluorometer (both Thermo Fisher Scientific, Waltham, MA, USA). A total of 750 ng of purified UHMW DNA was labeled with the Direct Label and Stain (DLS) kit (Bionano, #80005), and the concentration of the labeled DNA was measured with the dsDNA High-Sensitivity kit, again on a Qubit 2.0. Upon passing the QC, the DNA was loaded into G2.3 Saphyr Chips (#20366) and run on the Saphyr instrument to obtain 1300 Gbp with >75% map rate, in order to achieve sufficient genome mapping and >300× coverage per sample. The average UHMW DNA molecule size (N50 ≥ 150 kbp) was 289.5 kbp (+/− 41.2 kbp (std. dev.)).

### 2.3. Bionano Access/Rare Variant Analysis Pipeline

The rare variant analysis (RVA) pipeline was used for SV calling (Bionano Solve v3.7.1) against GRCh38 (Genome Reference Consortium Human Build 38) as the reference sequence, and the resultant SVs were visualized and extracted with the Bionano Access software v1.7.1. This bioinformatic approach targets somatic (≥5% allele fraction with 300× coverage) variants (“Rare”) alongside germline variants. Variant allele fraction (VAF) was modeled using a Bayesian approach and output as a measure of SV population clonality ranging from 0 to 1.

Data were subsequently filtered to remove the common variants observed in Bionano’s internal control database of 179 individuals. The human samples’ (hg38) SV calls were compared against the DGV (Database of Genomic Variants)’s supporting variants (release date 25 February 2020) obtained from http://dgv.tcag.ca/dgv/app/downloads (accessed 16 August 2023). Specifically, the criteria for the SV comparisons were the following: (1) insertion: any overlap with DGV gains; (2) deletion: position overlap by at least 50% of both Bionano calls and DGV loss calls (specified by the parameter ins_del_size_percent_similarity); (3) duplication: position overlap by at least 50% of both Bionano calls and DGV gain calls (specified by the parameter duplication_size_percent_similarity); and (4) inversion breakpoint: position overlap +/− 50 kb (specified by the parameter inversion_position_overlap).

### 2.4. SV Filtering and Identification of Disrupted Gene Loci

SVs for each patient sample were downloaded from Bionano Access after processing with the rare variant analysis pipeline (RVA, see above). Only the SVs detected in 0% of the Bionano control sample database were used in our downstream analysis. For enhanced stringency, only q-values (adjusted *p*-values) of <0.05 were deemed acceptable for drug–SV associations. The genes overlapping each SV were classified as disrupted based on the GRCh38 of the Human Genome Project.

### 2.5. Curve-Fitting of Drug Sensitivity Data

The cell viability results were used to compute the area under the curve (AUC) using the nplr package version 0.1-7 in R version 4.2.1. For the AUC computation, the following R function was first generated:            get_nplr = function(x, y) {
              tryCatch (nplr(x = x, y = y/100),
            error = function(e){ −1; })
}

The drug concentrations were converted from μM to M as follows:CONC <- Conc/1e6
and the AUC for each drug was computed as follows:nplr.res <- get_nplr (CONC, Viability)
    drug.auc <- unlist (getAUC(nplr.res)[1]) * 0.5

The “Best” (default) parameter was selected to test the models with different parameters and select the best goodness-of-fit model. The AUC for all the samples were consolidated and used for the downstream analysis. The drug target and pathway information was obtained from the Selleckchem L3000 Anti-cancer Compound Library information sheet (https://file.selleckchem.com/downloads/library/20201228-L3000-Anti-cancer-Compound-Library-96-well.xlsx (accessed 16 August 2023)).

### 2.6. Gene–Drug Interaction Significance Test

The gene–drug interactions were computed using Welch’s two-sample t-test in R. The 26 patient samples’ genes were categorized as “disrupted” if the given gene was disrupted by any SV type or as “wildtype”. Only the genes disrupted in two or more samples were considered for our analyses. A minimum of two samples were required for each of the disrupted and wildtype groups to be considered for significance testing, and only q-values (adjusted *p*-values) of <0.05 were deemed significant for rigor.

### 2.7. Drug Collections

Various iterations of the Oncology Dose Library (ODL, ODL2 and ODL3) were developed with the FDA-approved oncology collection drugs from the NCI’s Developmental Therapeutics Program (https://dtp.cancer.gov/dtpstandard/chemname/index.jsp (accessed 16 August 2023)). Experimental agents and those in phase II/III trials were acquired from Selleck Chemicals Inc. (Houston, TX, USA) or Cayman Chemical Company (Ann Arbor, MI, USA).

### 2.8. Cell Viability Assay and Drug Screening

Drug screening was essentially carried out as previously described [23,24,25]. Briefly, the leukemia-enriched cells were isolated as described above, and 2.5 k cells [in 25 μLs mTeSR1 Complete media (StemCell Tech., #85850, Vancouver, Canada)] per well was plated onto 384-well TC-treated cell culture plates (Greiner BioOne, #781098, Kremsmünster, Austria) previously spotted with 25 nLs of 1000× drugs with a Labcyte Echo 555 acoustic dispenser (Labcyte, San Jose, CA, USA). The final drug concentrations were 0.1, 1, and 10 μM. The cells were incubated with drugs for 96 h before cell viability was assessed indirectly by means of an ATP depletion luminescence assay with 10 μL per well of CellTiterGlo (Promega Corp., Madison, WI, USA) on an EnVision plate reader. The values were normalized to % viability by comparison to vehicle (0.1% DMSO)-treated control wells in Microsoft Excel (Version 2311 Build 16.0.17029.20140).

### 2.9. Data Sharing Statement

De-identified patient data and clinical details are provided in Appendix A. All the detected SVs and the SV–AUC drug sensitivity data can be found in Appendix A, respectively. All the genes with changes in SV VAF in the samples post relapse are provided in Appendix A. For access to the original/raw data, contact dfinlay@sbpdiscovery.org.

## 3. Results and Discussion

### 3.1. Optical Genome Mapping Detects All Structural Variants That Had Been Observed via the Standard of Care and Multiple Additional Variants Not Previously Reported

The leukemia samples used in this study (see also [24]) were obtained from Scripps MD Anderson. For this pilot study, we initially analyzed 26 individual leukemia samples from 23 patients, 3 of whom also provided samples upon their relapse. The samples comprised diverse subtypes (the majority being AML samples but also including an acute lymphoblastic leukemia (ALL) sample and an acute biphenotypic leukemia (ABL) sample), and the samples had previously been analyzed by means of traditional cytogenetics, including FISH/karyotyping, and, in some cases, also via targeted mutational sequencing. All the samples were originally identified as AML before two of them were reclassified as ALL and ABL, based on karyotyping, and were confirmed using OGM. The clinical characteristics and traditional cytogenetic analysis of each patient sample and, where available, the targeted mutational sequencing results are presented in Figure 1, and the complete de-identified patient metadata are provided in Appendix A. As these data are de-identified, this is the totality of the patients’ characteristics available in this study.

The samples were subjected to SV analysis using optical genome mapping (OGM), and over 1300 Gbp (>300× coverage) of genomic structural data were obtained for each sample, as detailed in the Methods. Any SVs detected above threshold (set at 0%) in a database of population controls were removed (see Section 2), and, for this study, the data were further filtered to only include the SVs overlapping annotated genes. This resulted in the detection of multiple novel SVs in every patient leukemia sample tested. The Individual samples harbored anywhere from less than twenty to over forty such distinct highly filtered variants, with all the patient samples displaying insertions, deletions, duplications, CNVs, and/or inversions and, often, also translocations (Figure 1, and Appendix A).

Our cohort of samples represent several genetically simple and/or “normal” karyotypes and other more perturbed samples, as reported by traditional cytogenetic analyses (Figure 1). OGM detected all the variants that had originally been discovered through traditional cytogenetics with 100% concordance (Figure 2A), similar to other studies which have utilized OGM in hematological malignancies [16,26,27,28,29,30,31,32,33,34]. All the SVs overlapping protein-coding genes in at least two samples and at 0% frequency in a database of healthy volunteers are depicted in the oncoprint image in Figure 1, and a comprehensive list of all the detected structural variants is provided in Appendix A.

As an example of a novel finding with potential disease relevance, sample −009NR was obtained from a patient after relapse (NR, “non-responder”), and, although Chr.7 monosomy was known to exist upon the initial presentation, a translocation (t(2;11)) had not been reported in the original sample prior to relapse using traditional cytogenetics (Figure 2B). Interestingly, this translocation was not predicted to result in gene fusion; however, a t(2;11) translocation resulting in a *NUP98*::*ASH1L* gene fusion has been previously reported in AML and was in fact detected at the time using OGM [20]. The study by Tembrink et al. that describes this *NUP98*::*ASH1L* gene fusion further describes how sub-clonal SVs develop during AML disease progression [20]. Unfortunately, we do not have any remaining pre-treatment samples from this patient. Nevertheless, our finding suggests one of the following: traditional cytogenetic methodologies are insufficient for detecting this type of anomaly; the clonal frequency was too low for detection using traditional methods; and/or clonal evolution of the disease was responsible for the presence of such a genomic event in the relapsed sample. Indeed, this translocation was modeled to occur at a variant allele frequency (VAF) of 0.12: simplistically, it was present in ~12% of the observed −009NR DNA molecules covering these breakpoints. Furthermore, this leukemia sample showed a 1.49 Mbp deletion spanning the *FOXP1* gene locus in chromosome 3 (also including *EIF4E3*). This deletion is another clonal SV associated with a copy number loss value of ~1.65 over the same region, suggesting the presence of a sub-clonal population with heterozygous deletion of the *FOXP1* locus which accounts for ~17.5% of the sample analyzed (Figure 2C).

Another example of an SV having potential disease relevance is a single copy deletion of the *EBF1* gene in an ALL sample (−013) (Figure 2D). *EBF1* (Early B-Cell Factor 1) encodes a transcription factor known to regulate the expression of proteins associated with the differentiation of nucleated hematopoietic cells and, as such, could be expected to be functionally and clinically relevant in hematological malignancies. We note that the deletion of this locus (1.09 Mbp) also results in the disruption of a copy of the *RNF145* and *LOC101927697* genes; however, the potential significance of these genes to leukemia is unclear at this time.

In sum, these data strengthen the suggestions of several previous studies that OGM can be superior to traditional cytogenetics in uncovering both known and novel SVs [26,27,28,29,30,31,32,33,34] and further suggest that OGM is potentially capable of tracking patient disease progression and clonal variation [20,33].

### 3.2. Drug Sensitivities Are Associated with Novel Genomic Rearrangements

We next examined the potential association of SVs with drug sensitivity and/or resistance. All our samples were subjected to chemosensitivity analyses with 120 FDA-approved and late-stage investigational drugs (see Section 2). A heatmap of differential chemical sensitivities is depicted in Figure 3, and the data therein were used to generate an unbiased clustering relationship between the samples. We show the concordance of agents designed to inhibit the same targets (e.g., MEK inhibitors, mTOR inhibitors, microtubule disrupters, etc.) and also show that the nucleated cells derived from blood and marrow aspirates from the same patient (see samples −013, −016NR and −074) always cluster together based on drug sensitivities. Interestingly, in some cases, samples from the same patients obtained upon presentation and upon relapse (sample names bolded in Figure 3 for clarity) show dissimilar chemosensitivities (samples −057 and −075) and do not align together based on unbiased Euclidian clustering (Figure 3).

To further assess the relevance of SVs to drug activities in leukemia and to potentially identify SVs as biomarkers for predicting patient responsiveness to clinically relevant drugs, we provide an unbiased integrative association between these two disparate datasets. All the drugs showing a statistically relevant alteration of the response associated with an SV, overlapping a protein-coding gene, in two or more samples are depicted in a volcano plot in Figure 4. We categorized the different leukemia cases into “disrupted” or “wildtype” groups for each gene that is disrupted in at least two samples. We performed a systematic drug–gene interaction analysis by comparing the drug response results in the disrupted and wildtype groups using Student’s *t*-test. No sample was omitted from the study for any reason other than lack of the high-quality ultra-high molecular weight DNA required for OGM analysis. The total number of drug–SV interactions in at least two samples (and involving only SVs which had never been found in the healthy volunteer control database) was 3510, but only 183 of said interactions had a q-value (an adjusted *p*-value) of <0.05. Of those, 76 drug–SV interactions had a log2 foldchange (log2FC) difference between the wildtype and disrupted groups of >0.5 (for sensitization) or <−0.5 (for resistance). Finally, filtering for interactions that statistically associated with a cell viability of <70% in one population resulted in 61 relevant high-confidence drug–SV interactions. As noted earlier, only protein-coding canonical genes were chosen for the analyses and discussion in this study, but the drug–SV associations with all the detected genes are provided in Appendix A. Whilst multiple drug–SV associations were detected, we choose to focus next on some potentially clinically relevant discoveries for brevity.

### 3.3. BCR-ABL1 Translocated Leukemia Samples Are Sensitive to Nilotinib but Exhibit Resistance to Proteasome Inhibitors

As a “proof-of-concept” interrogation of our diverse integrated datasets, we chose the well-characterized *BCR-ABL1* fusion translocation [35] as an example. Figure 5A shows Circos plots from two patient leukemia samples previously diagnosed as having *BCR-ABL1* translocations identified using traditional cytogenetics and confirmed here using OGM (Figure 2A). These samples (−013 and −015) had a canonical *BCR-ABL* translocation detected using OGM (Figure 5A) but also demonstrated multiple other genomic rearrangement events (Figure 1).

Indeed, the *BCR-ABL1* translocated leukemia samples were shown to be more sensitive to the second-generation Abl kinase inhibitor Nilotinib than those without said translocation (Figure 5A, lower panels). In reverse, the *BCR-ABL1* translocated leukemia samples were found to be less sensitive to proteasome inhibition with the clinically relevant agent Carfilzomib. The fact that the *BCR-ABL1* translocated samples were sensitive to Nilotinib (and Dasatinib) was no surprise, as it had been developed to target the kinase activity of the *BCR-ABL1* oncogene. Rather, this unbiased proof-of-concept result demonstrates that pertinent drug sensitivities relevant to particular SVs can be revealed with the integrated approach used in this study.

Interestingly, a patient leukemia sample that showed chemosensitivities tending toward those of the *BCR-ABL1* translocation samples had a unique and hitherto undescribed duplication of *CNTNAP2*, the gene encoding the Contactin-associated protein-like 2 (Appendix A). *CNTNAP2*, similar to other members of the neurexin family, possesses multiple epidermal growth factor repeats. Thus, speculatively, this duplication could result in the increased activation of downstream signaling events similar to those taking place in the *BCR-ABL1* translocated samples. Such pathway analysis studies will be the focus of future investigations as we expand the cohort of leukemia samples analyzed.

### 3.4. Drug Sensitivities and Resistance of Leukemia Samples with KMT2A Variants

The *KMT2A* gene encodes histone-lysine N-methyltransferase 2A, sometimes also referred to as mixed-lineage leukemia 1 (MLL1). Genetic rearrangements of this gene have been previously associated with both AML and ALL subtypes of leukemia, and, interestingly, the presence of *KMT2A* translocations is actually associated with low overall mutation rates [36]. Here, we show the detection of a *KMT2A-MLLT3* fusion translocation in patient leukemia samples and show that said rearrangement is associated with overall sensitivities to the closely related drugs Paclitaxel and Cabazitaxel (Figure 5B). This unexpected association with sensitivity to microtubule disruptors is conversely paralleled by a resistance to the clinically relevant class of Bcl-2 family inhibitors (ABT-737 and Navitoclax in our drug collection, Figure 5B, lower panels). Indeed, a close analogue of these agents, Venetoclax, is widely used clinically for multiple leukemia subtypes [37,38].

Notably, several of our samples lacking the *KMT2A-MLLT3* translocation instead possessed an insertion event in the *KMT2A* gene (Figure 5C). Manual inspection, however, suggested that this event is likely a partial tandem duplication (PTD). This notion is supported by recent OGM evidence in triple-negative breast cancer, where it was shown that some “insertion calls were probably mostly TD too small to be called as TD by RVA” [39]. Rather than showing a chemosensitivity profile similar to those of the translocation event, however, these samples were more sensitive to targeted kinase inhibitors, including Volasertib, a PLK1 inhibitor with orphan drug status in AML [40]. Furthermore, these samples were exquisitely sensitive to three structurally related nucleoside analogues (Figure 5C, lower panels), one of which, Cytarabine, is a frontline agent in certain forms of leukemia, including AML [41]. Thus, these results suggest that perturbations of the *KMT2A* gene are associated with changes in chemosensitivity involving clinically relevant drugs.

### 3.5. Unbiased Chemosensitivity Associations with SVs Unveil Potential New Treatment Strategies for Leukemia

Our systematic unbiased analysis of drug–gene interactions identified further novel biomarkers of drug sensitivity in our sample cohort of patients with leukemia. For example, we noted altered chemosensitivity to the “-’rubicin” family of closely related anthracyclines. As mentioned above, the DNA-damaging agent Idarubicin is often a frontline AML therapeutic in combination with Cytarabine [41] (Figure 5C, above). Idarubicin and its close chemical analogues, Daunorubicin and Epirubicin, were shown to be more effective against the leukemia samples with an insertion in the gene *MUSK*, encoding muscle-associated receptor tyrosine kinase (Figure 6A). The fact that these closely related chemical agents were unbiasedly associated with the same SV again suggests that our approach for personalized drug treatments associated with particular genomic variants could be meaningful. A further example concerns the duplication of the *FAM242D/FAM27C* gene locus, which is statistically associated with sensitivity to select kinase inhibitors Foretinib (c-Met and VEGFR-2), Sapanisertib/INK128 (mTOR), and Trametinib and PD325901 (both MEK inhibitors) (Appendix A).

As a final pertinent example, a gene locus that contains the Alkaline Phosphatase gene family is of particular interest (Figure 6B). A duplication of this genomic region correlates with an increased responsiveness to Teniposide and Fludarabine (and the EGFR inhibitor Dacomitinib) but with an increased resistance to the kinase inhibitors Ponatinib and Trametinib (Figure 6B, lower panels). Remarkably, the duplication of this gene locus was absent or below the detection threshold in one patient sample upon presentation (−075) but became apparent upon relapse (−075NR), correlating directly with sensitivity to the drugs mentioned above only in the relapsed sample. Curiously, this SV appeared to be a double duplication in one sample (−074) but a standard (single) duplication in the −075NR sample. We noted that this locus also contained non-protein-coding genes and pseudogenes, the potential significance of which is unknown at this point.

A more comprehensive assessment of the changes of the detected SVs from the samples upon presentation compared to those from the same patients upon relapse is provided through analyses of variant allele frequency, i.e., VAF. Figure 7 shows the plots of the changes of such SV VAF from three primary samples to the samples from the same patient upon relapse. Several SVs meeting our strict criteria were lost upon relapse whilst others were acquired. It is important to note here that the RVA pipeline calls gene duplications’ VAFs ~/= 1.0 due to the duplicated number of DNA molecule reads. These values have been retained unadulterated in Figure 7, just as they are called by the software (Bionano Solve v3.7.1), for the purposes of scientific data integrity; however, they likely represent heterozygotic duplications in actuality (blue spots). For all the SVs there was usually a cloud of VAFs of ~0.4–~0.6, which could be functionally thought of as VAF = 0.5 heterozygotes. There are several reasons why VAFs may differ from the theoretical values for a given set of criteria. Explanations of this include low clonality or true mosaicism for VAF << 0.5 or, potentially, a partial loss of heterozygosity for VAF >> 0.5 but <1.0 (e.g., Figure 7. Such alterations in clonality will be the focus of our future analyses.

In sum, we have demonstrated that the genome-wide detection of SVs using OGM [14,15,16,17,18,19,20,21,22,26,27,28,29,30,31,32,33,34] can be associated with predictions of sensitivity or resistance to various oncology drugs and that this approach could be a valuable biomarker identification system of potential clinical relevance. Due to the ex vivo nature of our studies, our drug response studies are limited to agents acting on the tumor itself. Accordingly, immunotherapies, such as CAR-T approaches, bifunctional antibodies, and T-cell engagers, that have been successful in the treatment of B-cell malignancies and that are still undergoing clinical testing for AML are beyond the scope of our studies. As such, we decided to focus on FDA-approved small molecules with the hope of identifying novel SV biomarkers of clinically relevant agents currently applied in AML but also to potentially identify drugs that have already been approved for other conditions that could potentially be re-purposed for patients with leukemia. We believe that such findings could be important, for example, for groups of patients that are not responsive to current standard chemotherapy and who are not eligible for clinical trials for various reasons.

In this regard, we note that an insertion in the *MUSK* gene was found to be associated with increased sensitivity to Idarubicin, while partial tandem duplication events in the KMT2A gene were related to the efficacy of Cytarabine. Our observations with Bcl2 family inhibitor drugs are also relevant. While the clinically approved Bcl2 inhibitor Venetoclax (ABT-199) was not present in our drug collections when this study began, two closely related homologues, Navitoclax (ABT-262) and ABT-737, were tested herein. Both these Bcl2 family inhibitors were found to be statistically associated with a reduced response in the samples with a MLLT3-KMT2A translocation (Figure 5B), and the Navitoclax/MLLT3-KMT2A translocation association was the greatest fold change in sensitivity of any of the interactions detected (Appendix A).

Inhibitors of IDH1 (Ivosidenib in our collection), IDH2 (Enasidenib in our collection), and FLT3 (Quizartinib in our collection) have been approved for AML, and their clinical use is targeted towards patients with specific mutations, as determined using FDA-approved companion diagnostic tests. Interestingly, we found no statistical association between these inhibitors and any of the SVs detected using OGM. In fact, although some of the AML samples tested showed some sensitivity to Quizartinib (Appendix A), no apparent pattern related to either FLT3 internal tandem duplication (FLT3-ITD) or FLT3 tyrosine kinase domain (FLT3-TKD) mutation (Figure 1) was apparent. There was little activity of Ivosidenib or Enasidenib against any of the samples tested (Appendix A), and, unfortunately, our sample set is lacking information on the mutation status of IDH1 and IDH2. Upon the expansion of our sample pool in the future, it will be of interest to assess whether certain SVs might associate with drug resistance.

Finally, the data we present confirm previous studies’ findings that suggested the utility of OGM for assessments of disease progression and increased pathogenesis [20,33]. Although traditional bulk sequencing implies a stepwise progression of malignant diseases, it cannot elucidate disease evolution or the associated clonal architecture therein [20]. Our future studies are intended to expand upon these possibilities and further ascertain differences in SVs and chemosensitivities in pre-treatment and relapsed patient samples for rigorous biomarker discovery.

## Figures and Tables

**Figure 1 cancers-16-00418-f001:**
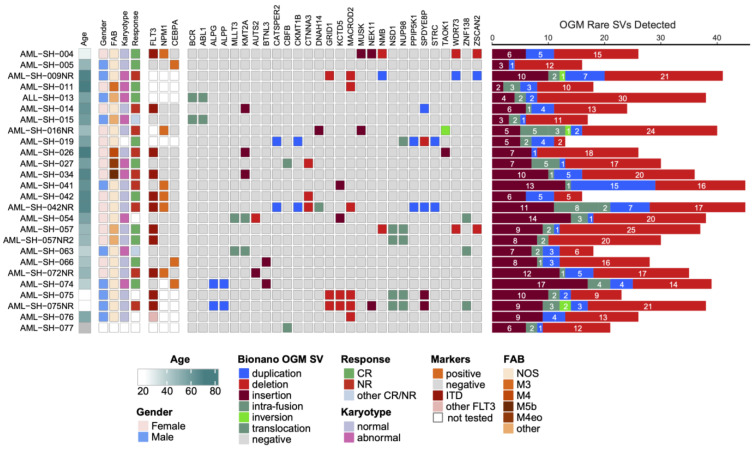
Oncoprint schematic of the characteristics, cytogenetics, and targeted mutational analyses for the 26 leukemia samples included in this study. FAB denotes a French–American–British classification. As for the response, NR is a non-responder, and CR is a complete responder. Positive cytogenetics and marker results indicate detected mutations, while negative results indicate a wildtype. The SVs overlapping protein-coding genes and never detected in healthy volunteers are also displayed. SV color-coding is maintained throughout the manuscript for clarity. Burgundy, insertion; bright red, deletion; green, inversion; blue, duplication; and teal, intra-fusions/translocations. For the response category, “other” signifies “morphologic remission and MRD-positive” for −015 and “initial NR upon induction (7 + 3) but CR upon treatment with fludarabine, then mitoxantrone + etoposide” for −063. The right side bar graph depicts the total numbers of the highly filtered SVs detected in each of the patient samples using OGM.

**Figure 2 cancers-16-00418-f002:**
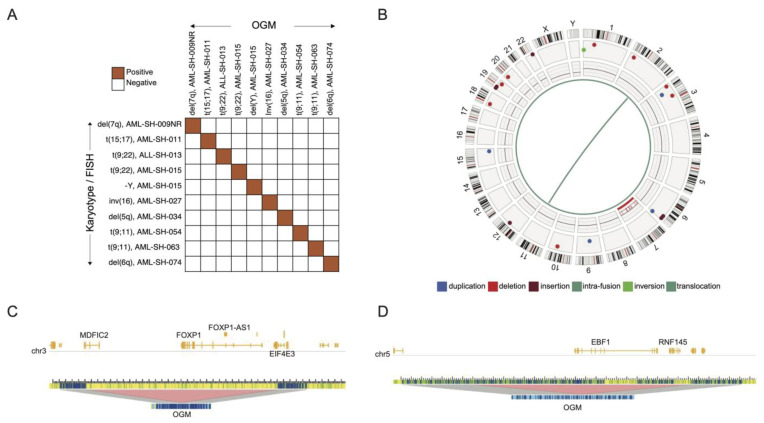
Summary of the patient samples’ optical genome mapping (OGM) of structural variants (SV). (**A**) The comparison of structural variants detected using cytogenetic methods and OGM shows a perfect concordance of the results. (**B**) Circos plot of SVs detected using OGM in a non-responder patient sample, AML-SH-009NR. The various panels (outside to inside) denote human chromosomes with cytobanding, canonical hg38-annotated genes, detected SVs, gene track/copy number variations, and translocations (inside). The various SV types are color-coded as described above and as used in Figure 1. Genome browser view of deletion SVs at the (**C**) *FOXP1* and *EBF1* loci showing an example of a sub-clonal population with heterozygous deletion (*FOXP1*) in −009NR and (**D**) a heterozygous deletion event (*EBF1* gene locus) in −013.

**Figure 3 cancers-16-00418-f003:**
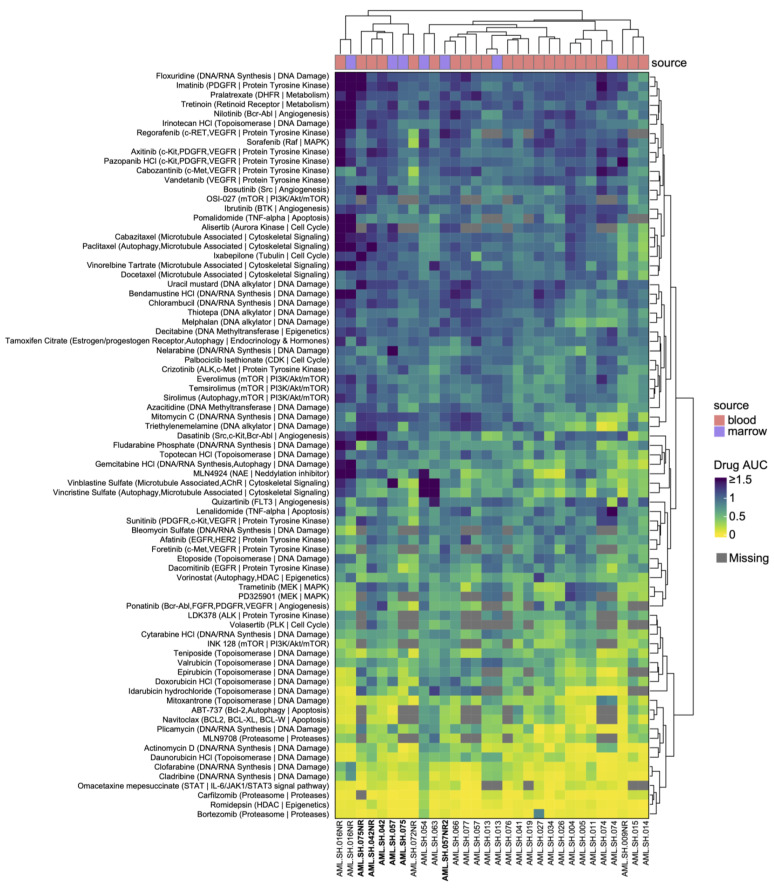
Systematic screening of drugs from the Oncology Drug Library (ODL) to identify markers of therapeutic sensitivity and resistance. Heatmap of the area under the curve (AUC) values for ODL (Oncology Drug Library) drugs screened against the patient samples. Only the drugs exhibiting a significant response (AUC ≤ 0.6) in at least one sample are shown. The molecular targets of the drugs or their associated pathways are also shown. The source of the samples is indicated using red (peripheral blood) or cyan (bone marrow aspirate), showing concordance in the drug responses of bone marrow and blood samples from the same patient where available (042 and 042NR, 057 and 057NR2, and 075 and 075NR).

**Figure 4 cancers-16-00418-f004:**
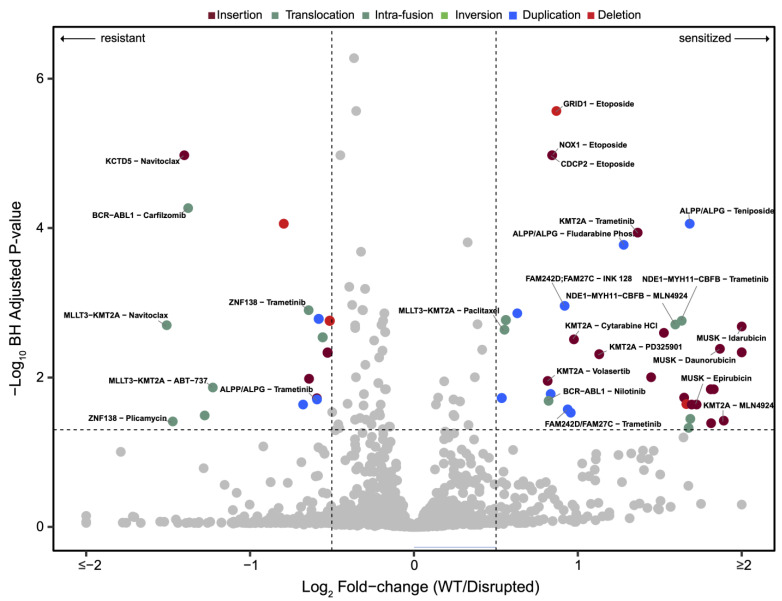
Volcano plot showing the magnitude and significance for unbiased drug–SV interactions. Each circle represents a drug–SV interaction. Significant interactions are denoted with colors corresponding to the SV type. Selected protein-coding drug–SV interactions discussed in the main text are labelled. The horizontal and vertical lines represent the significance thresholds of adjusted *p*-value (<0.05 and see Section 2.6) and AUC fold change (1.5-fold), respectively. The SV types disrupting each gene in the “disrupted” samples are denoted using the same color codes as Figure 1.

**Figure 5 cancers-16-00418-f005:**
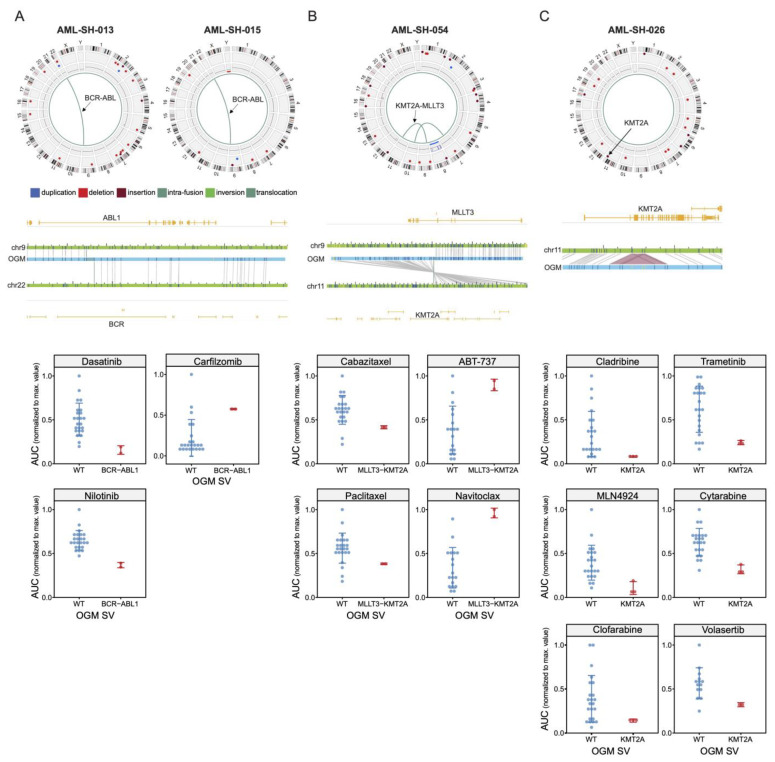
Known leukemic SV events are associated with sensitivity or resistance to clinically relevant drugs. (**A**) (**Upper panels**) Circos plots denoting t(9;2) *BCR-ABL1* translocations in samples ALL-SH-013 and ABL-SH-015, as determined using OGM. (**Middle panels**) Gene-level map showing the translocation breakpoint. (**Lower panels**) Drug sensitivity plots to Dasatinib, Nilotinib, and Carfilzomib of the relative AUC values of the WT vs. *BCR-ABL1* translocated samples. (**B**) (**Upper panel**) Circos plot example from a sample with a t(9,11) *MLLT3-KMT2A* translocation. (**Middle panels**) Gene-level map depicting the *MLLT3-KMT2A* gene fusion breakpoint. (**Lower panels**) Drug sensitivity scatter plots (for Cabazitaxel, Paclitaxel, ABT-737, and Navitoclax) of the relative AUC values of the WT vs. *MLLT3-KMT2A* translocated samples. (**C**) (**Upper panel**) Circos plot example from a sample with a *KMT2A* partial tandem duplication SV (Chr. 11, arrow). (**Middle panels**) Gene-level map depicting said *KMT2A* partial tandem duplication SV and its location. (**Lower panels**) Drug sensitivity plots of the relative AUC values of the WT vs. *KMT2A* partial tandem duplicated samples.

**Figure 6 cancers-16-00418-f006:**
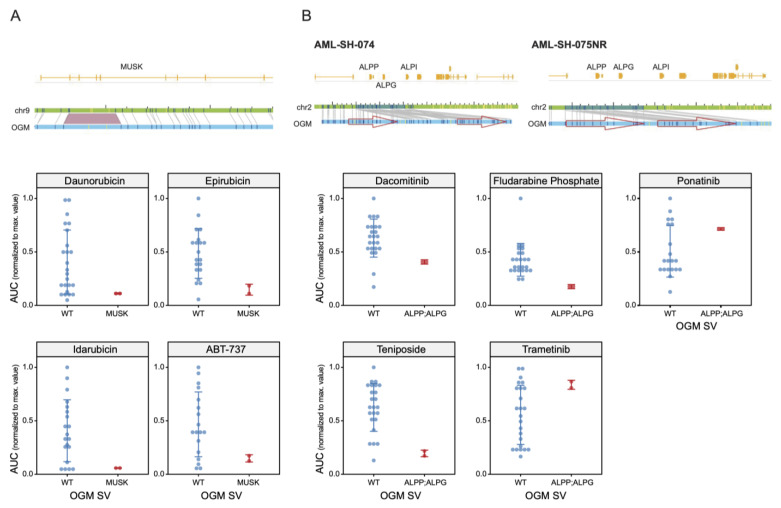
Identification of potential SV biomarkers of clinically relevant drugs. (**A**) (**Upper panels**) Genome browser view of an insertion event in the gene *MUSK* associated with sensitivity to the clinically relevant “-’rubicin” family of anthracyclines. (**Lower panels**) Scatter plots showing that the leukemia cases with an SV disrupting the gene *MUSK* (above) are more sensitive to Daunorubicin, Epirubicin, and Idarubicin. (**B**) (**Upper panel**) Genome browser views of duplication events in the Alkaline Phosphatase (ALP) family gene locus. (**Lower panels**) Drug sensitivity plots of the relative AUC values of the WT vs. *ALPP*, *ALPG*, and *ALPI* duplication SVs depicted above.

**Figure 7 cancers-16-00418-f007:**
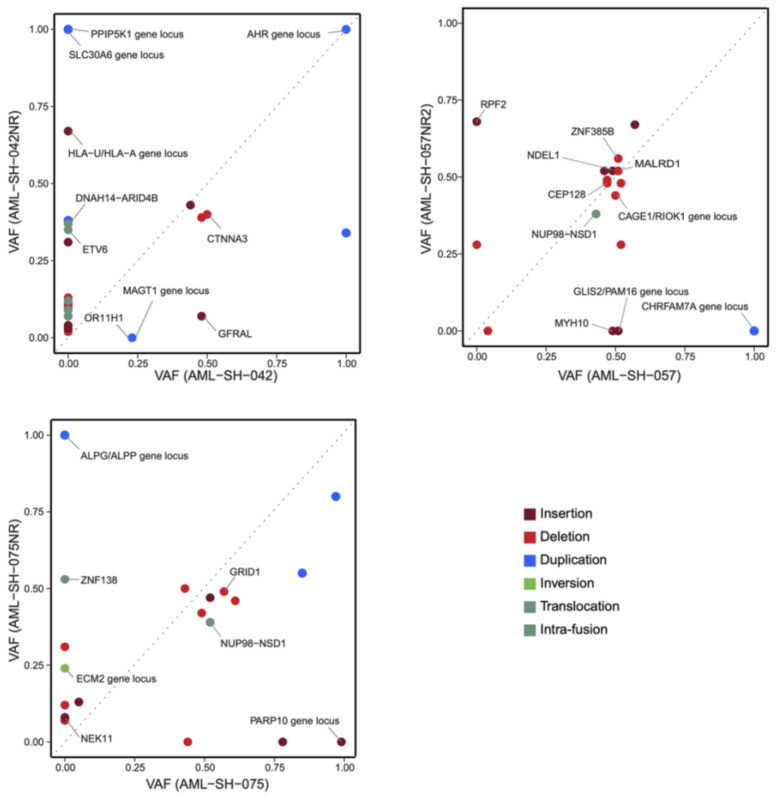
Plots of changes in SV variant allele fractions (VAF) during disease progression. VAF differences from a sample upon presentation (x-axis) vs. after relapse (y-axis) for samples 042, 057, and 075.

## Data Availability

The data presented in this study are available in this article (and Appendix A).

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
