# Peer review of "Detection of Genomic Structural Variations Associated with Drug Sensitivity and Resistance in Acute Leukemia"

_cancers, 2024, doi:10.3390/cancers16020418_

Round 1

Reviewer 1 Report

Comments and Suggestions for Authors

Despite of the relatively low number of samples authors managed to obtain important and significant data. It seems that the use of optical genome mapping may be of great clinical significance about drug sensitivity and resistance of patients with AML. I highly appreciate the accuracy and in time handling of the clinical samples.

My evaluation of the manuscript is highly positive and I do not have critical remarks with the exception that the authors could provide some more detailed information about the patients characteristics. 

Author Response

Reviewer 1:

Despite of the relatively low number of samples authors managed to obtain important and significant data. It seems that the use of optical genome mapping may be of great clinical significance about drug sensitivity and resistance of patients with AML. I highly appreciate the accuracy and in time handling of the clinical samples.

My evaluation of the manuscript is highly positive and I do not have critical remarks with the exception that the authors could provide some more detailed information about the patients characteristics. 

We are delighted the reviewer assessed our study so positively. Unfortunately, only the details provided in Supp. Table 1 are available to us. As the patient characteristics have been de-identified. We have updated the text (section 3.1) to more clearly state this.

Reviewer 2 Report

Comments and Suggestions for Authors

This manuscript demonstrates that structural variations detected by optical genome mapping (frozen samples from patients with acute leukemia) can be used to forecast drug sensitivity: a few but clinically relevant gene-drug pairs  (e-g- MUSK-idarubicin, KMT2A/MLL-citaravin, blc2-i) have been supported.

Personalized clinical trials can be exploited base don the above findings.

Author Response

Reviewer 2:

This manuscript demonstrates that structural variations detected by optical genome mapping (frozen samples from patients with acute leukemia) can be used to forecast drug sensitivity: a few but clinically relevant gene-drug pairs  (e-g- MUSK-idarubicin, KMT2A/MLL-citaravin, blc2-i) have been supported.

Personalized clinical trials can be exploited based on the above findings.

We appreciate the positive comments by the reviewer and indeed look forward to these studies ultimately contributing to patient care.

Reviewer 3 Report

Comments and Suggestions for Authors

In this work the authors analyze genomic rearrangements and other Structural Variations (SVs) through the utilization of optical genome mapping (OGM) in leukemia patients’ samples. This technology enables the unbiased detection of novel SVs, which may be relevant to disease pathogenesis and/or drug resistance. The authors also present an integrative analysis that matches these diverse datasets to multiple drug sensitivity profiles by testing more than 100 FDA-approved oncology drugs. The authors demonstrated that an insertion in the MUSK gene is associated with increased sensitivity to idarubicin, while partial tandem duplication events in the KMT2A gene are related to the efficacy of cytarabine. The approach presented in this work is innovative and could be useful for the development of personalized medicine.

Author Response

Reviewer 3:

In this work the authors analyze genomic rearrangements and other Structural Variations (SVs) through the utilization of optical genome mapping (OGM) in leukemia patients’ samples. This technology enables the unbiased detection of novel SVs, which may be relevant to disease pathogenesis and/or drug resistance. The authors also present an integrative analysis that matches these diverse datasets to multiple drug sensitivity profiles by testing more than 100 FDA-approved oncology drugs. The authors demonstrated that an insertion in the MUSK gene is associated with increased sensitivity to idarubicin, while partial tandem duplication events in the KMT2A gene are related to the efficacy of cytarabine. The approach presented in this work is innovative and could be useful for the development of personalized medicine.

Thank you for the positive comments. Opportunity to drive development of personalized medicine for patients continues to be a main motivation for our studies.

Reviewer 4 Report

Comments and Suggestions for Authors

The idea presented in the manuscript is interesting and the data presentation is quite intriguing. However, I have a few concerns related to the data and analysis, particularly in the second dataset focusing on Drug-SV interaction.

1.         Drug-SV Interaction:

a. It is imperative that the data for drug sensitivity, including cell viability assay and luminescence assay, be explicitly included in the manuscript. Not just AUC but viability data.

b. The correlation between this drug sensitivity data and SVs, and the subsequent inference regarding the role of SVs as biomarkers for drug sensitivity, requires further clarification. The linkage between SVs and drug sensitivity seems somewhat opaque.

c. I find the discussion of the implications of this data on AML speculative, with seemingly low relevance to current therapeutics. Could you provide more concrete evidence or context to support its significance?

d. How can the authors ascertain that the observed sensitivities, based on cell viability, are solely governed by SVs? A more detailed explanation or analysis may help in understanding this aspect better.

e. The relatively small dataset (n=26) with only one ALL and ABL sample might be susceptible to cherry-picking. How did you address this limitation in your analysis?

2.         Introduction:  The introduction should provide a more detailed rationale for associating chemical sensitivity with somatic SVs. Additionally, explaining how the correlation was calculated would enhance the clarity of your research. Also, a little detail about current drug therapies; specifically used in the analysis, for AML and ALL will be helpful.

3. Methods:  The cell viability based AUC calculation should be explained more. Relevant data should also be added to the results section.

4.         Figure 2:  It appears that the ALL 0013 sample is missing in Figure 2. Please ensure that all ALL samples are accurately represented in the figures.

5.         Wide Scale Application: a. Could you elaborate on the broader applications of your study? Do you believe that patients showing sensitivity to specific drugs, as indicated by your findings, could be considered for those drugs in a treatment setting? Elaborate in the discussion part.

6.         SV as a Biomarker: a. Is it possible that SVs could now be considered biomarkers for drug sensitivity? If so, could you provide insights into how this could be practically applied?

I believe that addressing these concerns would significantly enhance the clarity and robustness of your manuscript. I appreciate your efforts in advancing our understanding in this field and look forward to your response.

Author Response

Reviewer 4:

The idea presented in the manuscript is interesting and the data presentation is quite intriguing. However, I have a few concerns related to the data and analysis, particularly in the second dataset focusing on Drug-SV interaction.

We are thrilled the reviewer judged our manuscript so positively. We provide below a point-by-point response to the specific queries raised.

  1. Drug-SV Interaction:
  2. It is imperative that the data for drug sensitivity, including cell viability assay and luminescence assay, be explicitly included in the manuscript. Not just AUC but viability data.

We apologize for the original omission. The cell viability data are now provided in updated Supp. Table 3.

  1. The correlation between this drug sensitivity data and SVs, and the subsequent inference regarding the role of SVs as biomarkers for drug sensitivity, requires further clarification. The linkage between SVs and drug sensitivity seems somewhat opaque.

And similar to point (d) below:

Due to the novel datasets used for these integrative analyses we were very careful to apply extremely stringent statistical cut-offs for the unbiased results presented in the study. We had no a priori expectations for the interactions observed as the entire datasets were processed in an unbiased manner. For example, rather than utilizing p-values that could result in nebulous or opaque results due to the phenomenon of “p-hacking”, our bioinformatics specialists insisted on the use of adjusted p-values, also referred to as q-values, in order to weed out any potential erroneous associations. Furthermore, we applied various other “cut-offs” to ensure that any statistical associations were also robust enough to be of scientific, and potentially clinical, relevance. These parameters are detailed in the Methods (sections 2.3, 2.4 and 2.6), and in Results and Discussion (section 3.2), whilst Supp. Table 3 details the stratification of said associations which led ultimately to the rigorous associations presented in the manuscript.

  1. I find the discussion of the implications of this data on AML speculative, with seemingly low relevance to current therapeutics. Could you provide more concrete evidence or context to support its significance?

The reviewer is entirely correct that there have been many recent advances for the treatment of various hematological malignancies, most notably in the area of immunotherapies. In the case of AML, antibody-drug conjugates have been approved, while CAR-T approaches, bifunctional antibodies and T-cell engagers that have been successful in the treatment of B-cell malignancies are still in clinical testing for AML. Due to the ex vivo nature of our studies we are unable to test these intriguing modalities. As such, we decided to focus our studies here on FDA approved small molecules with the hope of identifying novel SV biomarkers to clinically relevant agents but also to potentially identify drugs that have already approved for other conditions that could potentially be re-purposed for leukemia patients. We believe such findings could be important, for example, for groups of patients that are not responsive to current standard of care and who are not eligible for clinical trials for various reasons. We have now updated the Results and Discussion section to present our rationale and findings in the context of current and emerging therapeutics (e.g. Bcl2, IDH1/2, and FLT3 inhibitors, and the modalities mentioned above), and thank the reviewer for pointing out our previous lack of discussion of these relevant points.

  1. How can the authors ascertain that the observed sensitivities, based on cell viability, are solely governed by SVs? A more detailed explanation or analysis may help in understanding this aspect better.

Please see response to similar point 1b above.

  1. The relatively small dataset (n=26) with only one ALL and ABL sample might be susceptible to cherry-picking. How did you address this limitation in your analysis?

Patient samples were originally collected for drug repurposing studies, and indeed we have drug screening data from many more samples. In this retrospective analysis, we only had sufficient material for OGM analyses from the samples described herein. No available sample has been omitted from the study for any reason other than lack of high quality ultra-high molecular weight DNA required for OGM analysis. We have updated the text (section 3.2) to stipulate this. Furthermore, all samples were originally thought to be AML upon presentation but were later re-classified as ALL and ABL based on karyotyping, and confirmed by OGM. We have further updated the text (section 3.1) to state this. A more detailed description of the statistical analyses/ cut-offs applied is described above in response to point 1b and 1d, above.

  1. Introduction: The introduction should provide a more detailed rationale for associating chemical sensitivity with somatic SVs. Additionally, explaining how the correlation was calculated would enhance the clarity of your research. Also, a little detail about current drug therapies; specifically used in the analysis, for AML and ALL will be helpful.

Similar to point 1c, please see our response above. We have updated the Introduction with a more detailed rationale, and the Methods section detailing the drug-SV association calculations has been updated, both as suggested by the reviewer.

  1. Methods:  The cell viability based AUC calculation should be explained more. Relevant data should also be added to the results section.

The original cell viability data is now provided in Supp. Table 3 (and discussed above in response to point 1a). The methodology for the AUC calculation is now provided in more detail in Methods section 2.5.

  1. Figure 2:  It appears that the ALL 0013 sample is missing in Figure 2. Please ensure that all ALL samples are accurately represented in the figures.

ALL-013 is present in Fig. 2. We are working with the publisher to provide them with higher resolution Figures so that the labelling is easier to read. We apologize for any confusion.

  1. Wide Scale Application: a. Could you elaborate on the broader applications of your study? Do you believe that patients showing sensitivity to specific drugs, as indicated by your findings, could be considered for those drugs in a treatment setting? Elaborate in the discussion part.

We thank the reviewer for this insight. We have now updated the Discussion section (and see response to point 1c above) to suggest potential application of our study in future. We note briefly here, however, that clinical application would require a clinical trial.

  1. SV as a Biomarker: a. Is it possible that SVs could now be considered biomarkers for drug sensitivity? If so, could you provide insights into how this could be practically applied?

I believe that addressing these concerns would significantly enhance the clarity and robustness of your manuscript. I appreciate your efforts in advancing our understanding in this field and look forward to your response.

It is our suggestion that at least some SVs could be considered biomarkers of potential therapeutic response. We suggest that SV detection by OGM will likely become routine, at least initially for hematological malignancies, and as such unbiased genome wide detection of even rare SVs will be possible. These data coupled with patient outcomes in a clinical setting could indeed result in more SVs being utilized as treatment biomarkers. We have added text to the Discussion to elaborate on the important points raised by the reviewer and believe it has greatly strengthened the manuscript.

In sum we again thank all the reviewers for their insight and comments and believe that based on their suggested improvements the paper has been strengthened and is now suitable for publication in Cancers.